 **eLIFE**

# A tethered delivery mechanism explains the catalytic action of a microtubule polymerase

Pelin Ayaz[1,2], Sarah Munyoki[1,2], Elisabeth A Geyer[1,2], Felipe-Andrés Piedra[1,2], Emily S Vu[1,2], Raquel Bromberg[1,2], Zbyszek Otwinowski[1,2], Nick V Grishin[1,2,3], Chad A Brautigam[1], Luke M Rice[1,2]*

[1]Department of Biophysics, UT Southwestern Medical Center, Dallas, United States; [2]Department of Biochemistry, UT Southwestern Medical Center, Dallas, United States; [3]Howard Hughes Medical Institute, UT Southwestern Medical Center, Dallas, United States

**Abstract** Stu2p/XMAP215 proteins are essential microtubule polymerases that use multiple αβ-tubulin-interacting TOG domains to bind microtubule plus ends and catalyze fast microtubule growth. We report here the structure of the TOG2 domain from Stu2p bound to yeast αβ-tubulin. Like TOG1, TOG2 binds selectively to a fully 'curved' conformation of αβ-tubulin, incompatible with a microtubule lattice. We also show that TOG1-TOG2 binds non-cooperatively to two αβ-tubulins. Preferential interactions between TOGs and fully curved αβ-tubulin that cannot exist elsewhere in the microtubule explain how these polymerases localize to the extreme microtubule end. We propose that these polymerases promote elongation because their linked TOG domains concentrate unpolymerized αβ-tubulin near curved subunits already bound at the microtubule end. This tethering model can explain catalyst-like behavior and also predicts that the polymerase action changes the configuration of the microtubule end.

*For correspondence: Luke. Rice@UTSouthwestern.edu

Competing interests: The authors declare that no competing interests exist.

## Introduction

Microtubules are dynamic polymers of αβ-tubulin that have critical roles in chromosome segregation and intracellular organization (reviewed in *Desai and Mitchison, 1997*). The polymerization dynamics of microtubules are regulated by multiple cellular factors. Evolutionarily conserved proteins in the Stu2p/XMAP215 family (*Gard and Kirschner, 1987*; *Ohkura et al., 1988*; *Wang and Huffaker, 1997*) regulate microtubule dynamics by promoting fast microtubule elongation. These essential proteins use multiple αβ-tubulin binding tumor overexpressed gene (TOG) domains to selectively recognize the growing microtubule end and promote its elongation (*Al-Bassam et al., 2006*; *Brouhard et al., 2008*; *Widlund et al., 2011*; *Al-Bassam et al., 2012*).

A significant advance in understanding occurred with a landmark study of XMAP215 in which in vitro reconstitution assays demonstrated that the polymerase affected the *rate* of microtubule elongation without affecting the apparent *equilibrium* (*Brouhard et al., 2008*). This and other observations (e.g., *Shirasu-Hiza et al., 2003*; *van Breugel et al., 2003*) formed the basis for describing XMAP215 as a catalyst for microtubule elongation. Catalytic action in turn led to the concept that TOG-containing polymerases might stabilize an otherwise rate-limiting intermediate along the polymerization pathway (*Brouhard et al., 2008*). However, an understanding of mechanism has been limited because the nature of this intermediate, and how TOG domains might selectively promote it, remained unclear.

A recent study from our group revealed that the TOG1 domain from Stu2p binds preferentially to a 'curved' conformation of αβ-tubulin that cannot be incorporated into the body of the microtubule

**eLife digest** Dynamic filaments of proteins, called microtubules, have several important roles inside cells. Microtubules provide structural support for the cell; they help to pull chromosomes apart during cell division; and they guide the trafficking of proteins and molecules across the cell.

The building blocks of microtubules are proteins called αβ-tubulin, which are continually added to and removed from the ends of a microtubule, causing it to grow and shrink. Other proteins that interact with the microtubules can help to speed up these construction and deconstruction processes. Ayaz et al. took a closer look at the structure of one particular family of proteins that make it easier for the microtubules to grow, using a technique called X-ray crystallography. The resulting images show two sites—called TOG1 and TOG2—on the enzymes that attach to the αβ-tubulin proteins. Ayaz et al. found that this binding can only occur when αβ-tubulin has a curved shape, which only happens when the tubulins are not included in, or are only bound weakly to the end of, a microtubule.

Previous research suggested that the two binding sites might work together to provide 'scaffolding' that stabilizes the microtubule. However, genetic experiments by Ayaz et al. show that microtubules will grow even if one of the binding sites is missing. Both TOG1 and TOG2 bind to αβ-tubulin in the same way, and by using computer simulations Ayaz et al. found that this helps to speed up the growth of microtubules. This is because the enzyme's two sites concentrate the individual tubulin building blocks at the ends of the filament. For example, TOG2 could bind to the end of the microtubule, while TOG1 holds an αβ-tubulin protein nearby and ready to bind to the filament's end. This tethering allows the microtubules to be assembled more efficiently.

(*Ayaz et al., 2012*). Our study also showed that a TOG1-TOG2 construct could bind two αβ-tubulins (*Ayaz et al., 2012*). This latter observation suggested that two TOG domains might cooperate to stabilize an αβ-tubulin:αβ-tubulin interface, and consequently that cooperative binding to αβ-tubulin might contribute to polymerase activity. Our study did not determine how the polymerase recognizes the extreme microtubule end, but we speculated based on apparent biochemical differences between the TOG1 and TOG2 domains (*Al-Bassam et al., 2006*; *Ayaz et al., 2012*) that selective interactions between TOG2 and a different, end-specific conformation of αβ-tubulin might be important.

In the present study we sought to gain insight into the mechanism of end recognition by determining the conformation of αβ-tubulin recognized by TOG2 and by testing whether cooperative TOG:αβ-tubulin interactions contributed to polymerase activity. We first determined the crystal structure of a TOG2:αβ-tubulin complex. This structure reveals that TOG2 binds to the same curved conformation of αβ-tubulin that TOG1 does. Our biochemical experiments underscore this structural similarity by demonstrating that the two TOG domains have comparable affinities for αβ-tubulin, with $K_D$ ~100 nM. Next, we used analytical ultracentrifugation to show that in TOG1-TOG2 the two linked TOG domains bind non-cooperatively to two αβ-tubulins. Non-cooperative binding indicates that TOG1-TOG2 does not stabilize an αβ-tubulin:αβ-tubulin interface. Together with biochemical and genetic experiments, our results lead to a model that explains how the polymerase activity can emerge from the action of two tethered TOG domains that each bind independently to a conformation of αβ-tubulin that is incompatible with the microtubule lattice. We propose that the polymerase activity arises because linked TOG domains selectively increase the effective concentration of αβ-tubulin near weakly bound, curved αβ-tubulins already on the microtubule end. A computational realization of this model supports our proposal by recapitulating catalyst-like behavior. The model further suggests that the polymerase achieves its effect in part by transiently altering the configuration of the growing end.

## Results and discussion

### TOG2, like TOG1, binds curved αβ-tubulin

We previously showed that the TOG1 domain from Stu2p binds preferentially to a curved conformation of αβ-tubulin (*Ayaz et al., 2012*). However, TOG1 is dispensable for the plus-end binding of Stu2p (*Al-Bassam et al., 2006*), and because of apparent differences in the biochemical behavior of TOG1 and TOG2 (*Al-Bassam et al., 2006*; *Ayaz et al., 2012*) we speculated that TOG2 might bind to a

different, lattice-induced conformation of αβ-tubulin (*Ayaz et al., 2012*). We solved the crystal structure of a TOG2:αβ-tubulin complex (*Figure 1*) to resolve this ambiguity. The structure was determined by molecular replacement from crystals that diffracted anisotropically to a minimum Bragg spacing of 2.8 Å (*Table 1*). The final model has good geometry (*Table 1*; Molprobity [*Chen et al., 2010*] clash score 1.79; 95.6% favored residues in Ramachandran plot) and has been refined to an $R_{free}$ of 0.259 ($R_{work}$ = 0.217).

The structure of the TOG2:αβ-tubulin complex is remarkably similar to that of the TOG1:αβ-tubulin complex (*Figure 1*; *Ayaz et al., 2012*). Conserved residues like W341 and R519 on the tubulin-binding surface of TOG2 make very similar contacts with αβ-tubulin as their equivalents in TOG1 (W23 and R200, respectively) (*Figure 1A,B*). The structure of TOG2 bound to αβ-tubulin superimposes with the structure of 'free' TOG2 (*Slep and Vale, 2007*) with 0.7 Å rmsd over 237 Cα atoms (*Figure 1C*). Because the TOG2 domains pack differently in the 'bound' and 'free' crystals, the similarity in structure suggests that these TOG domains are rigid modules that do not change conformation when they bind αβ-tubulin. Even though the structure reported here was obtained from a new crystal form, the conformation of αβ-tubulin in complex with TOG2 is nearly identical to that seen for TOG1-bound αβ-tubulin (*Figure 1D*) (12.3° of curvature in the TOG2 complex vs 13.1° in the TOG1 complex), and is

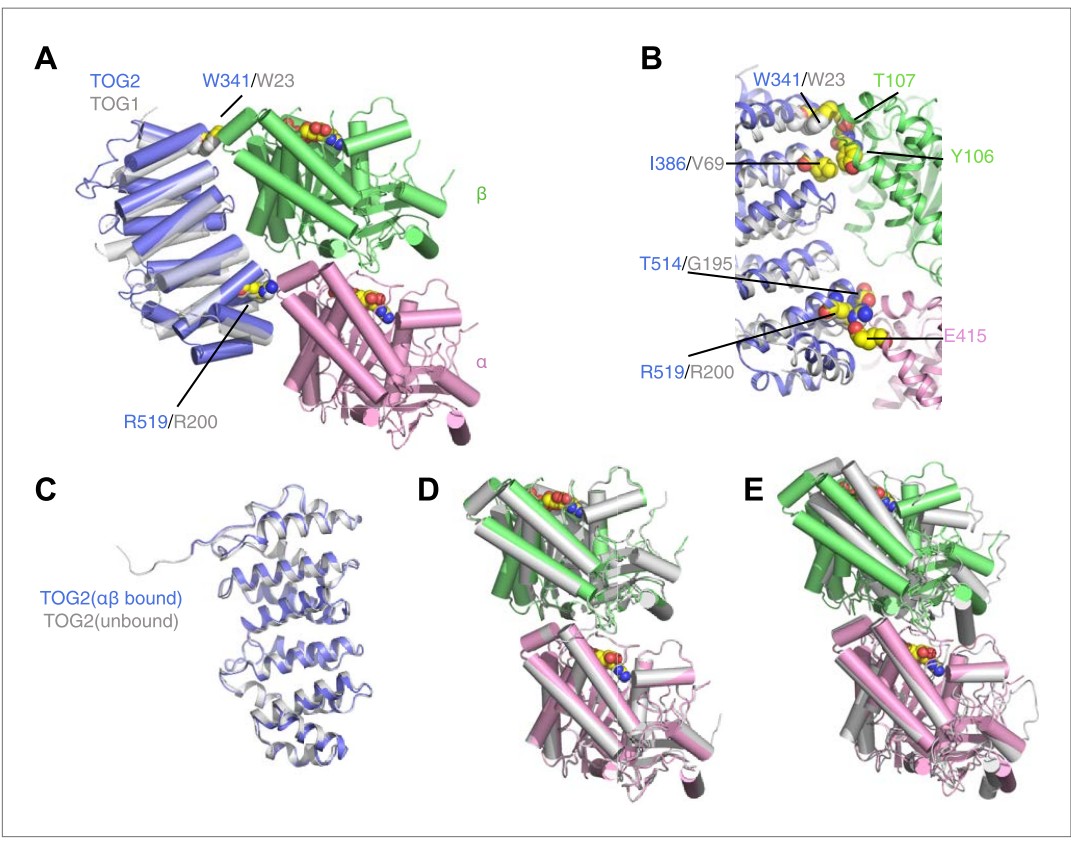

**Figure 1**. TOG2 binds to curved αβ-tubulin analogously to TOG1. (**A**) Structure of the TOG2:αβ-tubulin complex (TOG2: slate, α-tubulin: pink, β-tubulin: green), with the important binding residues W341 and R519 represented as spheres. The semi-transparent gray cartoon shows the previously observed binding mode of TOG1, with its binding residues W23 and R200 depicted as spheres. (**B**) Close-up of the TOG2:αβ-tubulin interface, colored as in **A**, and showing in spheres important interacting residues based on an earlier study. (**C**) Structural superposition of αβ-tubulin-bound (slate) and unbound (gray) TOG2 (PDB 2QK1). The two structures of TOG2 show only small local deviations, arguing against any significant conformational change associated with αβ-tubulin binding. (**D**) Structural superposition of the αβ-tubulin conformations in the TOG2 (colored) and TOG1 (gray) complexes. In both complexes αβ-tubulin adopts very similar curved conformations. (**E**) Structural superposition of TOG2-bound αβ-tubulin (colored) and the straight conformation of αβ-tubulin (PDB 1JFF, gray), showing the ~13° rotation of β-tubulin relative to α-tubulin that is characteristic of the curved conformation.

**Table 1.** Data collection and refinement statistics

| | |
|---|---|
| **Data collection** | |
| Space group | C2 |
| Cell dimensions | |
| a, b, c (Å) | 111.91, 89.57, 135.51 |
| $\beta$ (°) | 112.31 |
| Resolution (Å) | 50.0–2.81 (2.92–2.81)* |
| $R_{sym}$ | 0.143 (0.924) |
| <I>/<σI> | 9.6 (1.1) |
| Wilson B-value (Å) | 48.9 |
| Anisotropy (Å) relative to best direction (001) | |
| ΔB in (100) direction, ΔB in (010) direction | +29.95, +8.38 |
| CC1/2 in high resolution shell | 0.542 |
| Completeness (%) | 98.2 (91.3) |
| Redundancy | 4.1 (3.2) |
| **Refinement** | |
| Resolution (Å) | 2.81 |
| No. reflections | 26,235 |
| Completeness(%) | 86.5† (35.3) |
| $R_{work}$/ $R_{free}$ (%) | 21.8/25.9 (33.0/41.3) |
| Maximum likelihood estimated coordinate error (Å) | 0.42 |
| No. atoms | 8524 |
| Protein (non-hydrogen) | 8437 |
| Ligand/ion | 66 |
| Water | 21 |
| B-factors | |
| Protein | 44.7 |
| Ligand/ion | 52.0 |
| Water | 27.3 |
| Rms deviations | |
| Bond lengths (Å) | 0.003 |
| Bond angles (°) | 0.66 |
| Ramanchandran plot | |
| Favored (%) | 95.0 |
| Allowed (%) | 4.25 |
| Disallowed (%) | 0.75 |
| Rotamer outliers (%) | 3.2 |
| Molprobity clash score | 1.5 |

*Highest resolution shell is shown in parenthesis.
†The data were corrected for anisotropy in HKL2000. This treatment eliminated weak reflections and reduced the completeness of the data used for refinement compared to the completeness reported for data collection.

characteristically distinct from the straight conformation of αβ-tubulin (1° of curvature by our measure) (**Löwe et al., 2001**; **Figure 1E**). Underscoring this similarity, individual tubulin chains between the TOG1 and TOG2 complexes superimpose on each other with 0.4 Å rmsd over Cα atoms. Thus, both TOG1 and TOG2 bind preferentially to the same, curved conformation of αβ-tubulin. By extension, other TOGs in this family probably also bind to curved αβ-tubulin.

The shared preference of TOG1 and TOG2 for curved αβ-tubulin has implications for the mechanism of end recognition. Because TOG:αβ-tubulin interactions are required for plus-end localization of Stu2p/XMAP215 polymerases (**Al-Bassam et al., 2006**), preferential binding of TOGs to curved αβ-tubulin suggests that curved αβ-tubulin itself is the distinctive end-specific feature the polymerase recognizes. In contrast, other plus-end tracking proteins like Eb1 recognize lattice-specific features and consequently show 'comet-like' localization that extends into the microtubule body (**Nakamura et al., 2012**; **Maurer et al., 2014**). By invoking binding to an epitope that cannot exist in the body of the microtubule, our model explains how Stu2p/XMAP215 polymerases localize to the extreme microtubule end (**Nakamura et al., 2012**; **Maurer et al., 2014**).

The shared preference for fully curved αβ-tubulin also poses an apparent paradox, because it indicates that the polymerase is constructed from 'parts' that bind most strongly to a conformation of αβ-tubulin that cannot exist in the microtubule lattice. We explore and propose a resolution for this apparent contradiction in later sections.

## TOG1 and TOG2 each bind αβ-tubulin with comparable affinity

To determine if the structural similarity between the TOG1 and TOG2 complexes with αβ-tubulin extends to a biochemical similarity, we measured the affinity of TOG:αβ-tubulin interactions. Previously we had used polymerization-blocked αβ-tubulin mutants (**Johnson et al., 2011**) for TOG binding assays. To eliminate the possibility that blocking mutations and/or fluorescent labeling might mask or alter αβ-tubulin:αβ-tubulin interactions, we developed a label-free assay in which polymerization-competent αβ-tubulin could be used. Analytical ultracentrifugation monitored by absorbance at 230 nm (A230) allowed us to work at concentrations of αβ-tubulin at which higher-order oligomers of αβ-tubulin were nearly undetectable (**Figure 2A**). Indeed, at αβ-tubulin concentrations ranging from 0.08 to 1 µM there is

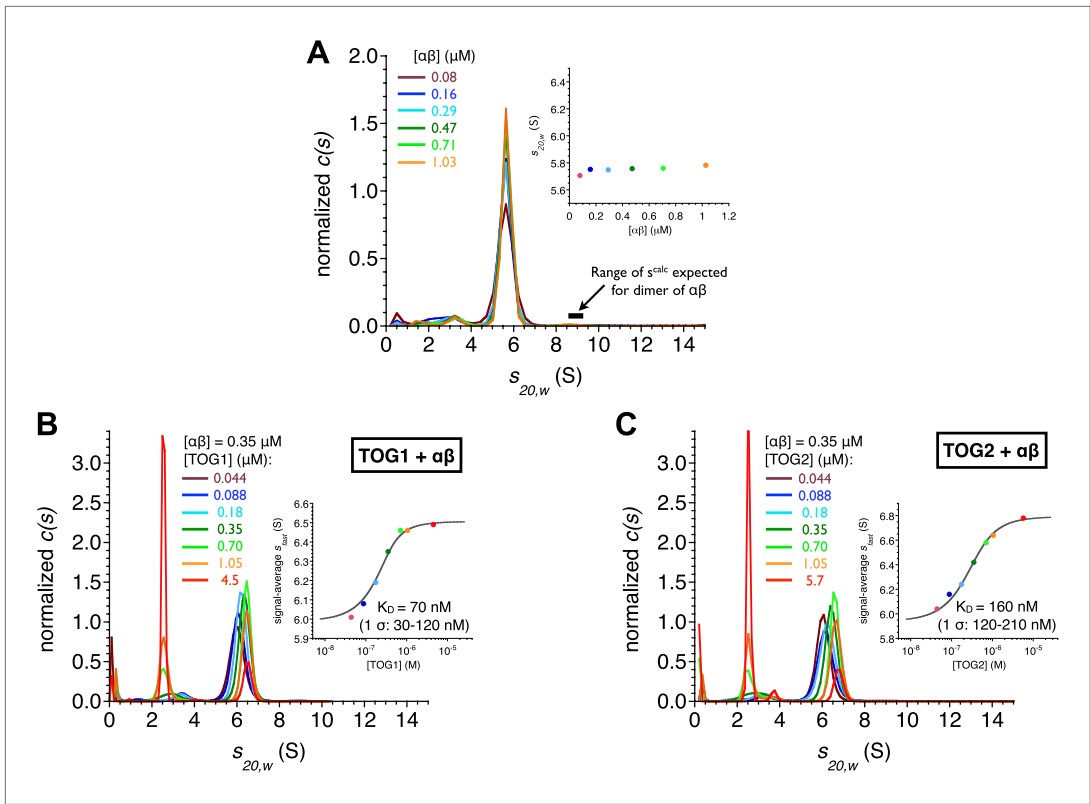

**Figure 2**. TOG1 and TOG2 bind αβ-tubulin with comparable affinity. (**A**) Sedimentation velocity analytical ultracentrifu-gation of polymerization competent yeast αβ-tubulin does not show signs of self-association between 80 nM and 1 μM concentration. The main plot shows c(s) distributions for a range of αβ-tubulin concentration. The inset shows that the $s_{20,w}$ is not increasing with αβ-tubulin concentration. Data points are color coded to match the c(s) distribution for that concentration. c(s), signal population as a function of s; $s_{20,w}$ sedimentation coefficient standardized to pure water and 20°C. (**B**) Analysis of TOG1:αβ-tubulin interactions by sedimentation velocity. The main plot shows c(s) distributions (color coded by concentration) for seven concentrations of TOG1 (44 nM–4.5 μM) titrated into 0.35 μM αβ-tubulin. The inset shows the fit (gray line) of a 1:1 binding isotherm to the signal average $s_{fast}$ (dots colored to match the c(s) distribution for that concentration of TOG1) resulting in a dissociation constant of 70 nM. (**C**) Analysis of TOG2:αβ-tubulin interactions by sedimentation velocity. Plots are as described in (**B**). The fitted dissociation constant is 160 nM.

a single dominant peak at 5.8 S with only 1.7 and 2.8% of the material sedimenting faster at the lowest and highest concentration tested. This uniform sedimentation behavior made it possible for us to use analytical ultracentrifugation as a quantitative binding assay.

We measured the affinity of TOG1 and TOG2 for αβ-tubulin by separately titrating variable amounts of each TOG domain into a constant amount of αβ-tubulin and analyzing the resulting sedimentation behavior (**Figure 2B,C**). We fit the concentration-dependent sedimentation profiles using single-site binding isotherms (**Dam and Schuck, 2005**; **Figure 2B,C**). The fits indicate that both TOG domains form relatively tight, 1:1 complexes with αβ-tubulin: TOG1 binds with $K_D$ = 70 nM and TOG2 binds with $K_D$ = 160 nM. The $K_D$ for the TOG2:αβ-tubulin complex reported here is consistent with our prior measurement using fluorescence anisotropy and polymerization blocked αβ-tubulin (**Ayaz et al., 2012**). Thus, in addition to a shared preference for the same curved conformation of αβ-tubulin, TOG1 and TOG2 also bind αβ-tubulin with comparable affinities. From both a biochemical and a structural perspective, the TOG1 and TOG2 domains are remarkably similar to each other.

## TOG1-TOG2 binds noncooperatively to two αβ-tubulins

When present together in a TOG1-TOG2 construct, each of the TOG domains can engage its own αβ-tubulin (**Ayaz et al., 2012**). We speculated that the two TOG-bound αβ-tubulins might also inter-act with each other, and that this cooperativity might make important contributions to polymerase function (**Figure 3A**).

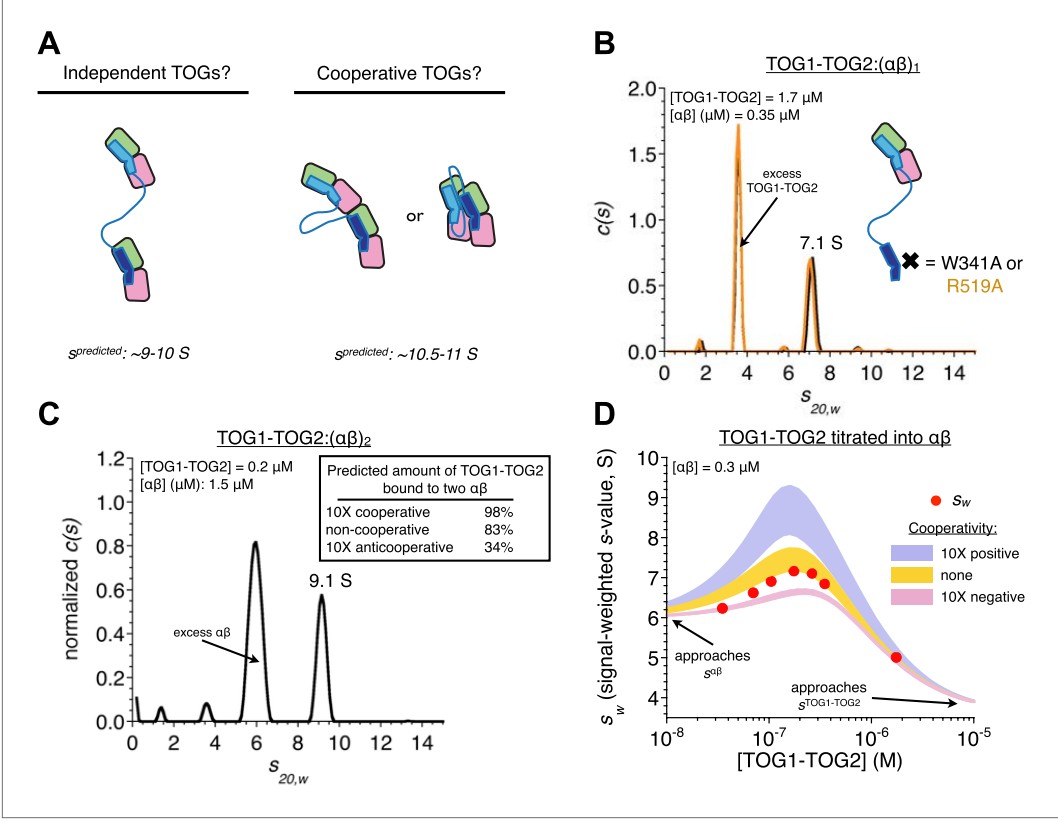

**Figure 3**. In TOG1-TOG2, the two TOG domains bind two αβ-tubulins without positive cooperativity. (**A**) Cartoons illustrating three different possible arrangements of a TOG1-TOG2:$(αβ)_2$ complex: independent (left) denotes that an αβ-tubulin:αβ-tubulin interface does not provide additional stability to the complex, cooperative (right) denotes that an αβ-tubulin:αβ-tubulin interface (either longitudinal or lateral) provides additional stability to the complex. Predicted sedimentation coefficients (calculated using HYDROPRO [**García De La Torre et al., 2000**]) are indicated. αβ-tubulin is represented in pink and green, TOG1-TOG2 in shades of blue. (**B**) Sedimentation behavior of a TOG1-TOG2:$(αβ)_1$ complex by sedimentation velocity AUC using two different mutations (W314A and R519A) that impair TOG2:αβ-tubulin interactions. The 'one tubulin' complex sediments at 7.1 S. (**C**) Placing limits on the sedimentation behavior of a TOG1-TOG2:$(αβ)_2$ complex by sedimentation velocity AUC. At ~5 molar equivalents of αβ-tubulin to TOG1-TOG2, the resulting complex sediments at 9.1 S. The inset shows the predicted fraction of TOG1-TOG2 engaged in 'two αβ-tubulin complex' under different assumptions about cooperativity. (**D**) Concentration dependence of TOG1-TOG2:αβ-tubulin interactions. Seven concentrations of TOG1-TOG2 were mixed with 0.3 μM αβ-tubulin and analyzed by sedimentation velocity AUC. Red dots indicate the signal-weighted $s_w$ values for the seven runs. The blue and pink swaths show the predicted behavior for TOG1 and TOG2 binding αβ-tubulin with 10-fold positive or negative cooperativity, respectively, and assuming the sedimentation coefficient of TOG1-TOG2:$(αβ)_2$ falls in the range 9.1–10.9 S (see text). The gold swath shows the predicted behavior for noncooperatively binding TOGs using the same range of sedimentation coefficient for TOG1-TOG2:$(αβ)_2$. The data are not consistent with cooperative binding of TOG1-TOG2 to two αβ-tubulins. Instead, they are much better described by independently binding TOG domains.

We used analytical ultracentrifugation to determine if cooperative interactions stabilized the TOG1-TOG2:(αβ-tubulin)$_2$ complex. First, using mutations on TOG1-TOG2 that disrupt the slightly weaker TOG2:αβ-tubulin interactions, we determined that TOG1-TOG2 bound to a single αβ-tubulin sediments at 7.1 S (**Figure 3B**). Next, using a mixture in which αβ-tubulin was superstoichiometric with respect to TOG1-TOG2 (**Figure 3C**), we observed a larger species sedimenting at 9.1 S. This value places a lower limit on the sedimentation coefficient for the TOG1-TOG2:(αβ-tubulin)$_2$ complex (**Figure 3C**, inset) that is consistent with hydrodynamic calculations (**García De La Torre et al., 2000**) using extended models in which the two αβ-tubulins do not contact each other and the TOG domains are separated by varying distances. Hydrodynamic calculations predict that compact models containing longitudinally or laterally of αβ-tubulins should sediment around 10.5–10.9 S (**Figure 3A,C**). Knowledge

about the sedimentation behavior of the one and two tubulin complexes allowed us to analyze a titration of TOG1-TOG2 into αβ-tubulin under different assumptions about cooperativity (*Figure 3D*). The resulting signal weighted average sedimentation coefficients are not consistent with models that assume even modest 10-fold positive or negative cooperativity (*Figure 3D*). Instead, the data are much better described by a model in which each TOG domain in TOG1-TOG2 interacts independently with its αβ-tubulin (the best fit was obtained from a model invoking less than twofold negative cooperativity, not shown). Noncooperative binding contradicts our initial expectation and indicates that αβ-tubulin:αβ-tubulin contacts do not provide additional stability to the TOG1-TOG2:(αβ-tubulin)$_2$ complex.

The lack of positive cooperativity stabilizing the TOG1-TOG2:(αβ-tubulin)$_2$ complex is striking because in this complex the two bound αβ-tubulins are physically constrained to occupy a relatively small volume and therefore are effectively at quite high concentration relative to each other. Indeed, allowing 55 Å for the length of each TOG domain and assuming that the ~75 amino acid linker is flexible and can maximally span 220 Å we estimate that the two TOG1-TOG2-bound αβ-tubulins are minimally at an effective concentration of roughly 200 µM, likely higher because the linker will rarely be fully extended. This effective concentration of TOG1-TOG2-bound αβ-tubulin is in the range of current estimates for the longitudinal $K_D$ for αβ-tubulins (e.g., *Gardner et al., 2011*). Observing noncooperative binding therefore suggests that some property of the TOG1-TOG2 linker antagonizes or counterbalances longitudinal interactions between αβ-tubulins (see below for experiments concerning the linker sequence). Our data do not rule out that linked TOG domains might stabilize lateral interactions between αβ-tubulins because these lateral interactions are thought to be much weaker (only molar affinity [*Gardner et al., 2011*]): even at effective αβ-tubulin concentrations approaching 1 mM a weak interface like this would not be populated and therefore would contribute little additional stability to the TOG1-TOG2:(αβ-tubulin)$_2$ complex. Whatever the underlying mechanism, the unexpected observation that the two linked TOG domains behave as independent αβ-tubulin binding modules places significant constraints on biochemical models for the mechanism of these polymerases.

## Two TOG domains are required for Stu2p function, but they do not have to be different

Stu2/XMAP215 family polymerases contain at least two different TOG domains. This property might indicate that the two domains have different functional specialization, but the structural and biochemical similarity of TOG1 and TOG2 (described above) does not seem consistent with separation of function. To investigate if different TOG domains are required for polymerase function we constructed a 'TOG-swapped' and other variants of Stu2p and assayed their ability to rescue the conditional depletion of endogenous Stu2p using a previously described assay (*Kosco et al., 2001*; *Al-Bassam et al., 2006*; *Ayaz et al., 2012*).

We first performed rescue assays using a 'full-length' Stu2p construct that retains the ability to dimerize and thus that contains two identical TOG1-TOG2 segments linked through the coiled-coil dimerization interface. A construct (TOG2-TOG2) in which TOG1 was replaced by a second copy of TOG2 rescued as well as did the wild-type (TOG1-TOG2) (*Figure 4A*). We also tried to replace TOG2 with a second copy of TOG1, but control experiments indicated that this variant was not stable (data not shown) and we have not yet pursued it further. Instead, we used site-directed mutagenesis as an alternate way to ablate the αβ-tubulin binding activity of TOG1 or TOG2. Constructs in which either TOG was impaired for αβ-tubulin binding (TOG1(R200A)-TOG2 or TOG1-TOG2(R519A)) also gave full rescue (*Figure 4A*). Ablating the function of individual TOGs is not without functional consequences under more stringent conditions, because these same constructs do show compromised rescue under conditions of microtubule stress (*Ayaz et al., 2012*). These data indicate that under normal conditions, the polymerase can function with only TOG1 or only TOG2 domains. Thus, having different TOG domains does not appear to be essential for polymerase function.

Using dimeric rescue constructs did not allow us to test Stu2p variants that contained only a single functional TOG domain of either kind. We therefore introduced the same mutations into a previously characterized dimerization-impaired variant of Stu2p in which the coiled-coil dimerization element had been deleted (Stu2p-Δcc) (*Al-Bassam et al., 2006*; *Figure 4B*). Consistent with prior observations (*Al-Bassam et al., 2006*), dimerization impaired Stu2p only partially compensated for the depletion of endogenous Stu2p. Even in this sensitized background, however, we found that as long as there were

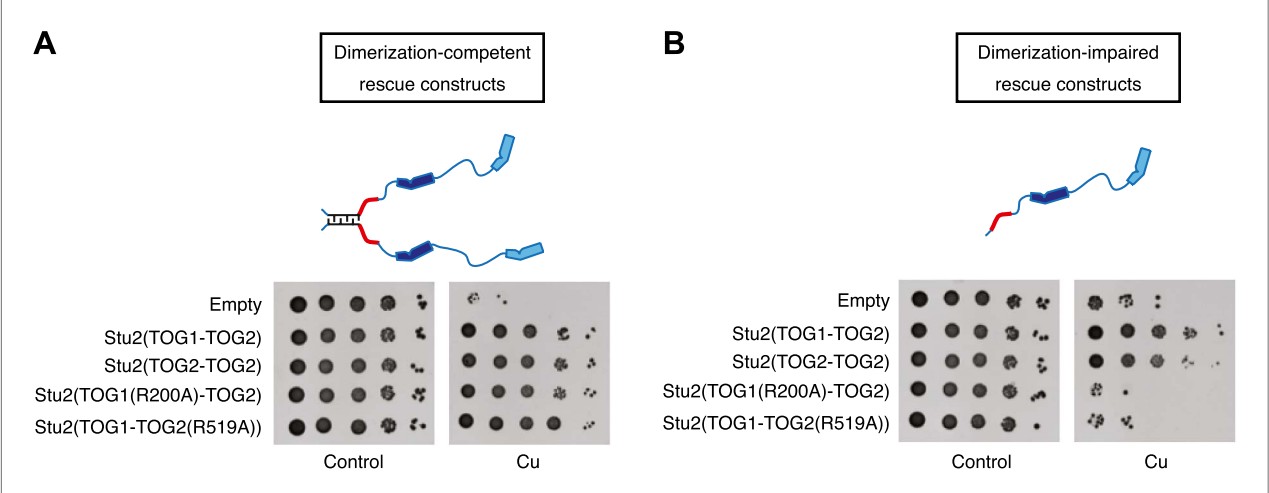

**Figure 4**. Two TOG domains are required for Stu2 function, but they do not have to be different. (**A**) Yeast carrying plasmid-based rescue constructs coding for dimerization-competent variants of Stu2p were plated at serial dilutions on media that was unmodified (control) or that contained 500 µM CuSO₄ (to deplete endogenous Stu2p; see text). All constructs, including those with debilitated TOG1 or TOG2 domains, showed full rescue. TOG domains are shown in blue, and the basic region in red. The coiled-coil is cartooned as a zipper. (**B**) As in **A** but using rescue constructs that are dimerization-impaired because the coiled-coil dimerization domain was deleted. In this more stringent background insults to either TOG domain abolished rescue activity. Replacing TOG1 with a second copy of TOG2 does not have adverse effects.

two functional TOG domains they did not need to be different. Stu2p(TOG2-TOG2)-Δcc showed rescue efficiency very similar to that of Stu2p(TOG1-TOG2)-Δcc (***Figure 4B***). Dimerization impaired variants of Stu2p in which TOG1 or TOG2 was defective for αβ-tubulin binding did not show any rescue activity (***Figure 4B***), consistent with a prior in vitro study of XMAP215 that demonstrated a requirement for at least two TOG domains (***Widlund et al., 2011***). More importantly, the ability of TOG2 to substitute for TOG1 in this more stringent, dimerization-impaired background strengthens the conclusion that the polymerase does not require different TOG domains for its function.

## Stu2p function tolerates substantial variation in the sequence linking two TOG domains

The dimeric Stu2p(TOG1-TOG2(R519A)) and Stu2p(TOG1(R200A)-TOG2) variants rescued the depletion of endogenous Stu2p in spite of the fact that the two functioning TOG domains were linked through the coiled-coil dimerization segment (and in the case of TOG1-TOG2(R519A) with a defective TOG domain in the linking sequence). This result suggested that how two TOG domains were linked was relatively unimportant, as long as they were linked. To explore this more systematically we determined how randomizing and/or shortening the TOG1-TOG2 linker in dimerization impaired Stu2p affected its rescue activity.

To test if the primary sequence of the TOG1-TOG2 linker was important for function, we prepared two variants of Stu2pΔcc in which the order of the central 65 amino acids of the TOG1-TOG2 linker (residues 252–316) was randomized (***Figure 5A***). This 'shuffling' strategy preserves the overall amino acid composition but should disrupt any local features specific to the natural sequence. Both shuffled linkers gave rescue activity nearly indistinguishable from the wild-type linker (***Figure 5A***), consistent with the robust rescue activity of alternatively linked functional TOGs in dimeric Stu2p(TOG1-TOG2(R519A)) and Stu2p(TOG1(R200A)-TOG2) (***Figure 4A***). Thus, the sequence linking the two functional TOG domains tolerates significant variation.

We also made a series of internal deletions in the TOG1-TOG2 linker to determine if shortening the linker affected rescue activity. Deleting 14 amino acids had little effect on rescue activity (***Figure 5B***). Rescue activity was mildly compromised by deletions of 22, 32, 40, and 50 amino acids (***Figure 5B***). Deleting 60 amino acids from the TOG1-TOG2 linker substantially abolished rescue activity (***Figure 5B***). These data suggest that Stu2p function is compromised when the sequence linking two TOG domains becomes too short. To examine the question of linker length more generally, we analyzed the distribution of TOG1-TOG2 linker length in ~300 Stu2p/XMAP215 orthologs. This analysis

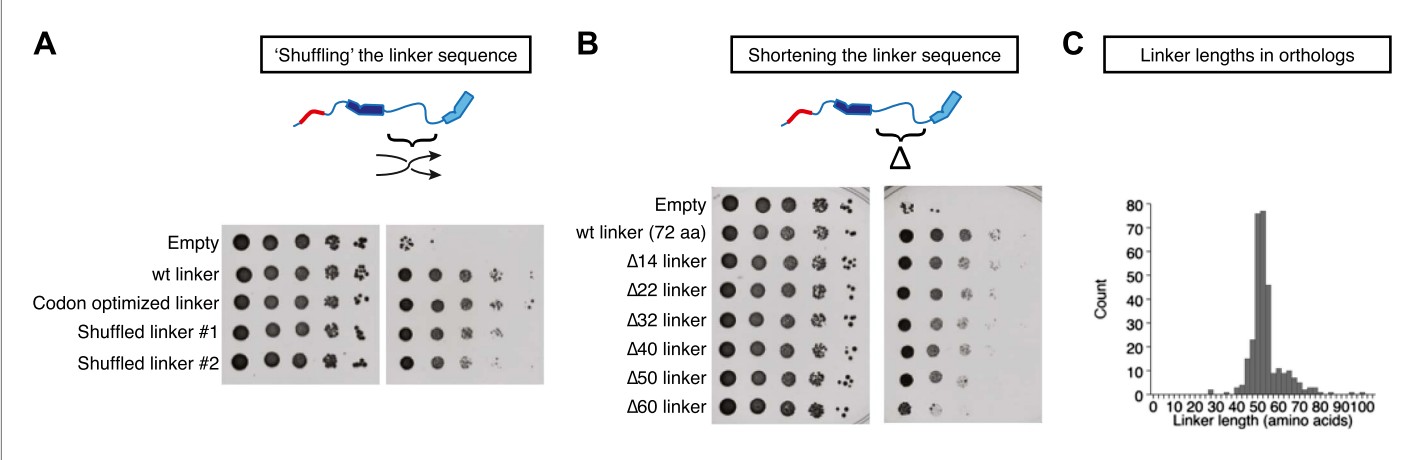

**Figure 5**. Stu2p function tolerates variation in the primary sequence and in the length of the TOG1-TOG2 linker. Rescue assays were performed as in *Figure 4*, using dimerization-impaired rescue constructs. (**A**) Stu2p variants with 'shuffled' (randomized) linker sequences rescue the depletion of endogenous Stu2p comparably to those with the natural linker. (**B**) Stu2p function is substantially abolished when the TOG1-TOG2 linker is truncated by 60 amino acids. Smaller truncations only show slightly compromised rescue activity. (**C**) Histogram illustrating the distribution of TOG1-TOG2 linker lengths in ~300 orthologs. The distribution shows that the linker length can vary but has a minimum tolerable length of ~40 amino acids.

showed that linker lengths below ~40 amino acids occur very rarely (**Figure 5C**), supporting the notion that there is a minimal linker length below which polymerase function is compromised.

## A simple 'tethering' mechanism

Any model for the polymerase function must recapitulate the catalyst-like properties documented previously (**Brouhard et al., 2008**) while also accounting for the seemingly paradoxical observation that both TOG1 and TOG2 from Stu2p bind preferentially to a curved conformation of αβ-tubulin that cannot exist in the body of the microtubule. A good model must likewise be consistent with the lack of detectable positive cooperativity stabilizing the TOG1-TOG2:(αβ-tubulin)$_2$ complex, and should not require different TOG domains or a non-spacer-like role for the sequence linking the two TOG domains.

In **Figure 6** we cartoon a simple 'tethering' mechanism that incorporates these new constraints, using a minimal 'two TOG' polymerase (**Widlund et al., 2011**) to simplify the representation. The central elements of this model are: (i) plus-end localization of the polymerase occurs by a TOG domain binding to curved αβ-tubulin at the microtubule end; (ii) full curvature of αβ-tubulin at the microtubule end requires that it not have any lateral neighbors; (iii) αβ-tubulin bound to the other TOG domain is physically tethered to the microtubule end and therefore associates with it more frequently. When not

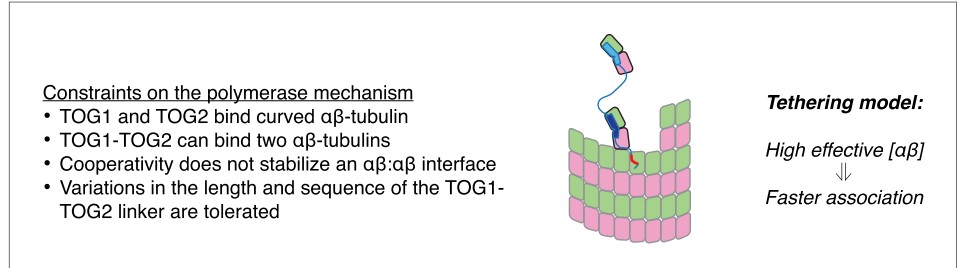

**Figure 6**. A tethering model for the polymerase function that incorporates our structural and biochemical observations. The model posits that MT plus-end recognition occurs through TOG-mediated recognition of curved (black outline, 'kinked' αβ-tubulin cartoon), not straight (gray outlines), αβ-tubulin on the MT end, and that the linked TOG domains (blue; the basic region is indicated in red) serve to 'tether' an unpolymerized αβ-tubulin to the MT end. Polymerase activity is predicted to arise because increasing the effective concentration of αβ-tubulin near the MT end should also increase the rate of αβ-tubulin:MT encounters.

engaged with αβ-tubulin at the microtubule end, we assume based on results from others (**Brouhard et al., 2008**) that the polymerase diffuses on the microtubule lattice by virtue of its basic domain, thereby rapidly 'searching' for another curved αβ-tubulin.

## An implicit kinetic model for the polymerase recapitulates observed aspects of polymerase activity

Can this tethered delivery model recapitulate the catalyst-like action of these polymerases? To address this question, we took advantage of a kinetic model for microtubule assembly that we have been developing as part of other ongoing work.

Our model for microtubule elongation is very similar to (and was inspired by) one previously developed by Odde and Cassimeris (**VanBuren et al., 2002**). Alternative and more complicated models exist (e.g., **VanBuren et al., 2005**; **Margolin et al., 2012**), but this relatively simple one captures the essence of the biochemistry—that the microtubule end presents a mix of high and low affinity binding sites, depending on neighbor context–in a reasonable way and using a small number of adjustable parameters ('Materials and methods'). Our model simulates microtubule polymerization one subunit association or dissociation at a time using kinetic Monte Carlo simulations on a two-dimensional lattice with staggered periodic boundary conditions to mimic the cylindrical structure of a 13 protofilament microtubule (**Figure 7A**). Because XMAP215 has been shown to promote the elongation of microtubules in the presence of a hydrolysis resistant GTP analog (**Brouhard et al., 2008**), we performed 'GTP-only' simulations (no GTP hydrolysis) to simplify the behavior by eliminating catastrophe. We performed a manual grid search of longitudinal and corner interactions (in the model these interactions are the dominant contributors to growth rate) to identify parameters that reproduced experimentally observed microtubule growth rates. To validate our program, we first fit the same experimental data (**Walker et al., 1988**) as did Odde and Cassimeris (**VanBuren et al., 2002**), obtaining similar parameters (**Figure 7—figure supplement 1**). Other measurements of microtubule dynamics (e.g., **Mitchison and Kirschner, 1984**; **Drechsel et al., 1992**; **Hyman et al., 1992**; **Brouhard et al., 2008**; **Gardner et al., 2011**]) have shown much lower critical concentrations than the Walker et al. data (**Walker et al., 1988**) do, so we also searched for parameters to approximate the growth rates observed in the presence of GMPCPP (**Brouhard et al., 2008**; **Gardner et al., 2011**). This procedure identified a different set of parameters (**Figure 7C**; **VanBuren et al., 2002**). These models provide a starting point for exploring potential polymerase mechanisms.

An explicit model for the polymerase requires too many adjustable parameters to be feasible at this time. To address the question 'can tethering give catalyst-like activity', we made a number of simplifying assumptions to create an implicit representation of polymerase action that isolates the 'tethered delivery' steps while ignoring other aspects of polymerase function and biochemistry. We assumed that every curved, longitudinally-bound 'singleton' αβ-tubulin at the MT end is engaged by one TOG domain of a polymerase (**Figure 7D**, left panel). This assumption provides end-localized polymerase in the model without having to parameterize/describe its diffusive movement on the microtubule lattice (this movement is mediated by the basic domain, which is required for polymerase function), or if it arrives at the microtubule end with its TOGs pre-loaded with αβ-tubulin (XMAP215 has been observed to carry αβ-tubulin while diffusing on the lattice). To mimic the tethering-induced local increase in αβ-tubulin concentration, in the model we accelerated the rate of αβ-tubulin association to the site neighboring the polymerase-bound singleton (**Figure 7D**, arrow). We do not assume that the linker is directing the tethered subunit to a particular site–separate simulations showed that accelerating the rate of αβ-tubulin association on top of the polymerase-bound singleton had no effect on growth rates (data not shown). We also assumed that when the polymerase bound singleton acquired a lateral neighbor it became straight (**Rice et al., 2008**; **Buey et al., 2006**), and that this straightening was linked to TOG disengagement (**Ayaz et al., 2012**).

To explore if enhanced trapping of singleton αβ-tubulins could recapitulate catalyst-like activity, we simulated microtubule elongation rates using a range of accelerated tethered association rates. In reality, the tethered association rate is likely to be more or less constant, with maximal polymerase activity instead determined by the degree to which it saturates its binding sites on the plus end. Varying the tethered association rate provides a way to mimic this 'saturation' effect without having to model it explicitly. Strikingly, the simulated microtubule elongation rates increase with the tethered αβ-tubulin association rate while the apparent equilibrium constant for elongation (x-intercept) remains approximately constant (**Figure 7D**, middle; **Figure 7—figure supplement 1**). Affecting the rate but

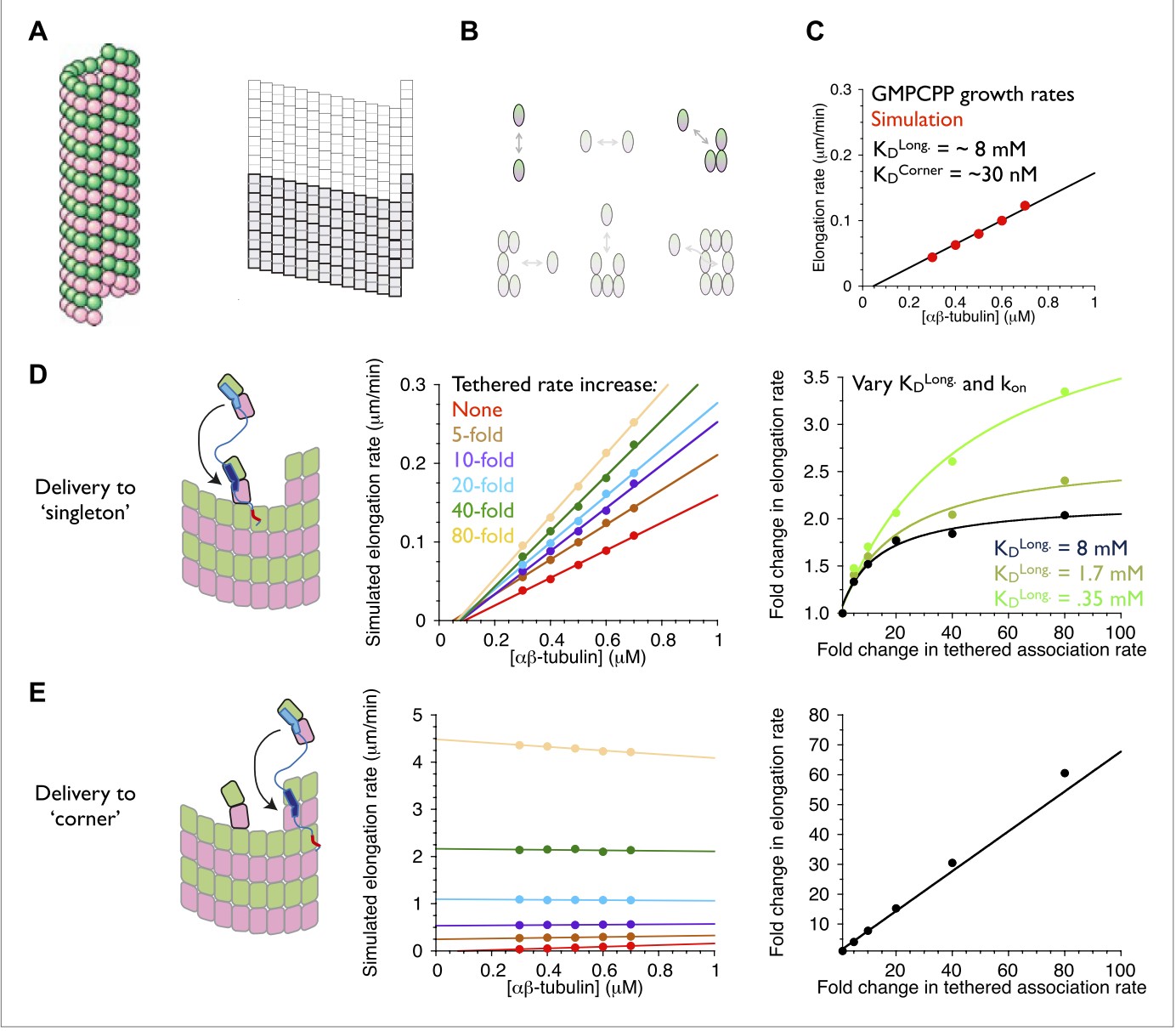

**Figure 7**. An implicit model can recapitulate the catalytic nature of polymerase activity. We developed a kinetic model for microtubule elongation and altered it to explore models for the polymerase. (**A**) Cartoon of the cylindrical microtubule (left; pink and green spheres represent α- and β-tubulin, respectively) alongside a two-dimensional representation of the MT lattice (gray boxes, right; the darker boxes represent the microtubule 'seed' used to template elongation in our simulations). (**B**) The model parameterizes all six possible neighbor states for αβ-tubulin in the lattice, but the two that dominate the elongation behavior are the longitudinal (top left) and corner (top right) interactions. (**C**) A grid search identified parameters capable of recapitulating the concentration dependence of microtubule elongation rates in the presence of GMPCPP. The black line summarizes the trend from experimental observations (***Brouhard et al., 2008***; ***Gardner et al., 2011***); red dots represent the results from our simulations. Using a $k_{on}$ of $4 \times 10^6$ $M^{-1}s^{-1}$, we obtained a good match to observed growth rates from $K_D^{long} = 8$ mM and $K_D^{corner} = 33$ nM. (**D**) (left) Cartoon illustrating the tethering model, with the polymerase (TOG domains in blue, basic region in red) localized to a curved αβ-tubulin bound at the MT end by pure longitudinal association, (middle) simulated growth rates obtained at increasing association rates for the tethered αβ-tubulin. Enhanced trapping of longitudinally-associated αβ-tubulin through the tethering effect shows catalyst-like activity: growth rates and apparent on-rate constant (slope) both change significantly but the apparent equilibrium for growth (x-intercept) does not. (right) Plot of fold increase in growth rate vs the fold increase in tethered αβ-tubulin association rate, using values at 0.5 µM αβ-tubulin concentration as a reference (black dots; the black line shows the fit of a hyperbolic curve to the growth rates). The model for the polymerase gives a relatively modest change in growth rates compared to the fold-increase in tethered association rate. Simulations with progressively stronger (dark green: 1.7 mM; light green, 0.35 mM) longitudinal interactions show higher maximal polymerase activity. The polymerase activity is related to the population of longitudinally-associated αβ-tubulin at the MT end. (**E**) Examining an alternative tethering model in which the

*Figure 7. Continued on next page*

*Figure 7. Continued*

polymerase promotes incorporation at a 'corner' site. This model yields much greater stimulation of elongation (middle) because there is always at least one corner site at the microtubule end. The predicted response also does not appear to saturate with increased tethering effect (right, linear fit). This alternative model does not describe the polymerase action because it fails to produce catalyst-like output.

The following figure supplement is available for figure 7:

**Figure supplement 1**. An implicit model as in **Figure 7** but trained against different measured growth rates.

not the apparent equilibrium indicates that tethering-enhanced associative trapping of weakly-bound singletons can recapitulate the essence of catalyst-like activity. The increase in simulated growth rate (less than 10-fold) is significantly less than the maximal increase in tethered αβ-tubulin association rate we explored (**Figure 7D**, right; **Figure 7—figure supplement 1**). Thus, in addition to capturing catalyst-like action, our simple model also shows the modest overall stimulation that is characteristic of these polymerases.

## Relationship between microtubule end structure and polymerase activity

Why did the model only stimulate elongation modestly, as the polymerases do? We speculated that because longitudinally associated αβ-tubulin is weakly/transiently bound at the microtubule end (~mM affinity in our fitted parameters and those of comparable models (**VanBuren et al., 2002**; **Gardner et al., 2011**); this compares to working concentrations of αβ-tubulin that are typically at most 20 μM), the low population and lifetime of this 'substrate' was limiting the activity obtainable in our model.

We probed the linkage between 'substrate scarcity' and polymerase activity in two ways. We first created an alternative model in which the polymerase accelerated delivery of αβ-tubulin adjacent to higher affinity and much longer-lived 'corner' (longitudinal + lateral interactions) sites, at least one of which is always present at the microtubule end (**Figure 7E**, left). In this alternative model the simulated elongation rates increase linearly with the increased effective concentration (**Figure 7E**, right), yielding much stronger stimulation that also lacks catalyst-like characteristics (**Figure 7E**, middle). This result is consistent with the idea that catalyst-like activity is related to the low affinity/lifetime of the end-bound singleton substrate. To explore this implication more directly we obtained parameter sets with higher affinity longitudinal interactions by using different assumed values of $k_{on}$. As observed previously (**VanBuren et al., 2002**), these alternative parameterizations can approximate the experimental rates of (uncatalyzed) microtubule elongation by compensating for faster/slower $k_{on}$ with weaker/stronger longitudinal affinity (but retaining the same 'corner' [longitudinal + lateral] affinity). We observed in simulations that stronger longitudinal affinity gave higher maximal polymerase activity (**Figure 7D**, right; see also **Figure 7—figure supplement 1**). These results support the notion that the polymerase activity in our model is related to the population/lifetime of longitudinally-associated αβ-tubulin singleton at the microtubule end.

The tethering model suggests a plausible transition state/collision complex that limits the rate of microtubule polymerization in the absence of a polymerase (**Figure 8**). The idea is that at the MT end, longitudinally-bound αβ-tubulins tend to dissociate much faster than the rate at which unpolymerized αβ-tubulins associate at neighboring sites. Accordingly, most of the 'singleton'-type associations are unproductive for elongation, and the MT grows primarily by additions into 'corner' sites. In the model we propose, Stu2p/XMAP215 polymerases enhance elongation by selectively concentrating αβ-tubulin near singleton sites, thereby specifically promoting their associative trapping. We propose that the 'trapping complex'—two side-by-side αβ-tubulins without additional neighbors to either side–may be the transition state. In our model the polymerase catalyzes formation of this 'side-by-side' state by selectively concentrating (tethering) the reactants that lead to it. Associative trapping of singletons also 'roughens' the microtubule end through the creation of additional, longer-lived 'corner' sites. By virtue of having more favorable sites of interaction, these rougher microtubule ends can transiently capture unpolymerized subunits more efficiently and independently of the polymerase, through 'filling in' (a contribution from filling-in was anticipated previously [**Brouhard et al., 2008**]).

Because our model does not explicitly represent the polymerase, it cannot explain the molecular mechanisms underlying processive action. Perhaps basic-domain mediated rapid diffusion on the microtubule lattice, as has been measured for XMAP215 (**Brouhard et al., 2008**), is sufficient for the liberated polymerase to engage a newly added αβ-tubulin.

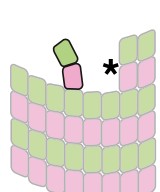

Curved, longitudinally bound αβ-tubulins are captured inefficiently; weak binding means that they dissociate rapidly compared to the rate of αβ-tubulin association. Efficient elongation mostly occurs from higher affinity 'corner' sites (*).

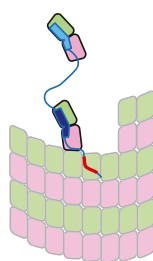

The polymerase binds to these weakly bound, curved αβ-tubulins at the microtubule end using one of its TOG domains. The other TOG domain can tether an unpolymerized αβ-tubulin, concentrating it near the polymerase binding site.

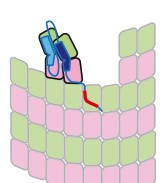

The tethering effect increases the likelihood that the weakly bound curved αβ-tubulin will be captured by adjacent association of the tethered αβ-tubulin. We propose that this 'side-by-side' complex is analogous to a transition state for elongation.

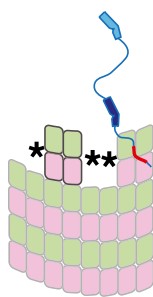

Acquisition of a lateral neighbor drives curved to straight conformational changes that result in polymerase release. The microtubule end also becomes 'rougher', with more high affinity 'corner' sites (*).

**Figure 8**. Schematic cartoons illustrating the origin of catalytic action. The microtubule end has multiple sites where αβ-tubulin can associate, but elongation is largely dominated by additions into the few, high-affinity 'corner' sites (left panel) because pure longitudinal associations are weak. By preferentially recognizing curved αβ-tubulin with one of its TOG domains, the polymerase (TOG domains in blue, basic region in red) can selectively localize to these 'unproductive' binding sites (second from left). The tethering action greatly enhances the rate at which these weakly bound subunits are trapped by neighboring association of another αβ-tubulin (middle two panels). Polymerization-induced straightening of αβ-tubulin releases the polymerase for another round of catalysis (right).

In summary, we propose that Stu2p/XMAP215 polymerases localize to the plus end by virtue of preferential interactions between TOG domains and curved αβ-tubulin. Our proposal is consistent with the view that curved, GTP-bound αβ-tubulin is an on-pathway polymerization intermediate (e.g., **Buey et al., 2006**; **Rice et al., 2008**; **Nawrotek et al., 2011**). It also explains the need for at least two TOG domains (**Widlund et al., 2011**) because one TOG is required to bind curved αβ-tubulin on the microtubule and another linked TOG is required for tethering an unpolymerized αβ-tubulin. In this model, side-by-side trapping and the release of TOG domains will be linked because of the polymerization-associated conformational changes that occur in αβ-tubulin. The model provides a simple molecular explanation for the catalytic-like action of the polymerase, recapitulates the modest maximal stimulation, and rationalizes how a polymerase can be built from domains that bind preferentially to a microtubule-incompatible conformation of αβ-tubulin.

## Materials and methods

### Plasmids and their construction

Plasmids to express TOG2 and wild-type or polymerization-blocked yeast αβ-tubulin have been described previously (**Johnson et al., 2011**; **Ayaz et al., 2012**) and were used without further modification. Some plasmids (to express 'full-length' Stu2p and its R200 and R519 mutants) for the rescue assay were also previously described (**Ayaz et al., 2012**). 'Dimerization-impaired' plasmids were constructed using mutagenesis primers to delete the region that codes for the coiled-coil dimerization element (corresponding to amino acids 658–761). These primers contained two regions that were complementary to the coding sequences just upstream and downstream of the deleted region. 'Linker truncation' plasmids were constructed using a similar strategy to remove increasingly large sections from the middle of the sequence linking TOG1 to TOG2 (deleted regions are: Δ14:273-286; Δ22:277-298; Δ32:273-304; Δ40:257-296; Δ50:251-300; Δ60:251-310). The 'linker shuffled' and 'TOG swapped' plasmids were made using GeneArt seamless cloning (Invitrogen, Carslbad, CA). Shuffled linkers were inserted starting from synthetic, codon-optimized cDNA (IDT, Coralville, IA), so we also prepared a codon optimized natural linker sequence to control for possible effects on expression levels. DNA coding for shuffled versions of the 65 central amino acids (residues 252–316) of the TOG1-TOG2 linker was obtained from IDT and amplified using primers to append flanking nucleotides of wild-type

upstream and downstream flanking sequence. The parent plasmid, excluding the region coding for the to-be-replaced linker sequence, was amplified in a separate reaction. The two products were mixed, incubated, and transformed according to the manufacturer's instructions. A similar strategy was used to replace TOG1 (residues 11–245) with a second copy of TOG2 (residues 318–560). The entire coding region of all plasmids was verified by DNA sequencing.

## Protein expression and purification

The TOG1(1–317), TOG2(318–560), and TOG1-TOG2(1–560) were expressed in bacteria with C-terminal His$_6$ tags and purified using Ni-affinity and ion exchange chromatography (*Ayaz et al., 2012*). Wild-type or polymerization-blocked yeast αβ-tubulin (β:T175R,V179R) were overexpressed in *Saccharomyces cerevisiae* (*Johnson et al., 2011*). All proteins were purified as previously described (*Johnson et al., 2011*; *Ayaz et al., 2012*).

## Crystallization, data collection, and structure determination

For crystallization, pure polymerization blocked yeast αβ-tubulin (β:T175R,V179R) and TOG2 domain were dialyzed into RB150 (25 mM Tris pH 7.5, 150 mM NaCl, 1 mM MgCl$_2$, 1 mM EGTA) and then mixed at equimolar stoichiometry. One tenth volume of 2 M L-proline was added before concentrating the protein using 30 kDa cutoff Amicon Ultra concentrators. When the desired concentration was reached (~2–3 mg/ml), GTP was added from a 100 mM stock to achieve a final concentration of 1 mM. Sparse-matrix crystallization screening (typically mixing protein with precipitants at 1:1 and 2:1 protein:reservoir ratios, using 200 nl of reservoir solution) was performed using a Phoenix DT Drop Setter (Rigaku, The Woodlands, TX). Needle- and blade-like crystals were obtained in multiple PEG-containing conditions, and these hits were optimized using finer screens. The crystal used for data collection was grown from 25% (vol/vol) PEG 3350, 0.25 M (NH$_4$)$_2$SO$_4$, 0.1 M MES pH 6.1). For harvesting, cryoprotectant (reservoir solution supplemented with 100 mM NaCl and 40% [vol/vol] glycerol) was added directly to the drop before looping crystals for freezing in liquid nitrogen in liquid nitrogen.

Diffraction data were collected at Argonne National Laboratory using APS beamline 19ID using remote data collection. Diffraction data were processed using HKL2000 (*Otwinowski and Minor, 1997*). Crystals adopt space group C2 with one complex in the asymmetric unit. The diffraction was anisotropic (*Table 1*). We used 2.81 Å as the high-resolution cut-off to avoid excessive loss of completeness.

All crystallographic calculations after diffraction data processing were performed using Phenix (*Adams et al., 2010*). Phases were obtained by molecular replacement, using as search models the yeast α- and β-tubulin chains from PDB 4FFB (*Ayaz et al., 2012*) with GTP removed, and the structure of TOG2 from Stu2p (PDB 2QK1 [*Slep and Vale, 2007*]). Model building was performed in Coot (*Emsley et al., 2010*). Disordered regions were removed, and GTP and new sidechains were manually placed where the electron density indicated that was appropriate. The model was refined conservatively, and based on the behavior of R$_{free}$ and other indicators we decided that a combination of TLS with two grouped B-factors for each residue represented an optimal strategy. We also ran tests to optimize the relative weighting of the X-ray and chemical restraints, consistently observing that optimal results (low R$_{free}$ and low R$_{free}$-R$_{work}$) were obtained when the covalent geometry was tightly restrained to ideal values.

## Analytical ultracentrifugation

Samples for analytical ultracentrifugation (TOG1, TOG2, TOG1-TOG2, and polymerization-competent yeast αβ-tubulin) were dialyzed into RB100 (25 mM Tris pH 7.5, 1 mM MgCl$_2$, 1 mM EGTA, 100 mM NaCl) containing 20 μM GTP. Samples were mixed and incubated at 4°C for at least one hour prior to the experiment. All analytical ultracentrifugation experiments were carried out in an Optima XL-I centrifuge using an An50-Ti rotor (Beckman–Coulter, Brea, CA). Approximately 390 μl of each sample were placed in charcoal-filled, dual-sector Epon centerpieces. Sedimentation (rotor speed: 50,000 rpm) was monitored using absorbance optics, and centrifugation was conducted at 20°C after the centrifugation rotor and cells had equilibrated at that temperature for at least 2.5 hr. Protein partial-specific volumes, buffer viscosities, and buffer densities were calculated using SEDNTERP (*Laue et al., 1992*).

The c(s) distributions were generated using SEDFIT (*Schuck, 2000*; *Schuck et al., 2002*). Isotherms for TOG1, TOG2, and TOG1-TOG2 binding to αβ-tubulin were assembled by importing c(s) distributions into GUSSI (available at http://biophysics.swmed.edu/MBR/software.html) and integrating them

to obtain the weighted-average sedimentation coefficients as a function of TOG concentration. These isotherms were exported to SEDPHAT (*Dam and Schuck, 2005*) for evaluation. For the binding of TOG1 and TOG2 to αβ-tubulin, a simple bimolecular binding model was used. For the binding of TOG1-TOG2 to αβ-tubulin, a two-site binding model (i.e., two αβ-tubulins binding to one TOG1-TOG2) with two microscopic association constants was used. Quantities measured in separate experiments (*s*-values of the individual components, *s*-values of TOG1-TOG2:(αβ-tubulin)$_1$, and the association constants derived with the single-domain TOG constructs) were treated as fixed parameters. We used a range of possible s-values for the TOG1-TOG2:(αβ-tubulin)$_2$, informed by different assumptions about cooperativity as shown in *Figure 5C* and by hydrodynamic calculations (*García De La Torre et al., 2000*). To calculate sedimentation coefficients, we first prepared compact 'longitudinal' and 'lateral' models by docking the TOG1 and TOG2 complexes onto the αβ-tubulins in the stathmin complex (pdb 1SA0 [*Ravelli et al., 2004*]) and a section of microtubule (coordinates provided by Ken Downing, Lawrence Berkeley lab), respectively. Extended models were generated by translating one TOG complex relative to the other. To calculate coordinates for the linker, we used the Modeller (*Sali and Blundell, 1993*) plugin to UCSF Chimera (*Pettersen et al., 2004*), calculating five different linker traces for each model in an attempt to capture some of the likely variability. Each resulting model was passed to HYDROPRO (*García De La Torre et al., 2000*), and sedimentation coefficients were calculated using a solvent density of 0.99823 g/ml and otherwise default settings.

## Genetic rescue assays

The genetic rescue assays were adapted from *Al-Bassam et al. (2006)* and performed as previously described (*Ayaz et al., 2012*). The parent strain was CUY1147ΔLEU2 (CUY1147 (*Kosco et al., 2001*) in which the LEU2 gene was replaced with a marker coding for G418 resistance). This strain expresses a tagged version of Stu2p that is selectively degraded upon exposure to copper. The parent plasmid was pWP70 (*Wang and Huffaker, 1997*), a CEN plasmid that expresses (non-degradable) HA-tagged Stu2p from its endogenous promoter. For rescue assays CUY1147ΔLEU2 transformed with rescue plasmid or empty vector was grown overnight in CSM-Leu media, normalized to A600 = 1, and plated at serial 10-fold dilutions onto CSM-Leu plates with or without 500 μM CuSO$_4$. Plates were incubated at 30°C or room temperature and imaged after several days.

## Evolutionary analysis of linker length

To collect a large number of TOG-domain containing sequences, we performed psiblast (*Altschul et al., 1997*) (non-redundant [nr] database, with e-value = 0.005 over three iterations) on amino acid sequences from TOG domains of known structure: the TOG1 and TOG2 domains from Stu2p (*S. cerevisiae*), the TOG2 domain from XMAP215 (*Xenopus laevis*) (*Slep and Vale, 2007*), and the TOG3 domain from Zyg9 (*Caenorhabditis elegans*) (*Al-Bassam et al., 2007*). The full protein sequences for all hits from each of the four runs were combined and identical hits were removed. They were then clustered using CLANS (*Frickey and Lupas, 2004*) (default parameters) over 130 iterations. The Stu2 cluster was extracted and aligned using mafft (*Katoh and Standley, 2013*) and passed to Jalview (*Waterhouse et al., 2009*). Proteins that were missing residues in highly conserved regions of TOG1 and TOG2, in particular in the boundaries adjacent to the start and end of the linker, were manually pruned. Based on the structures, we defined the boundaries for the end of TOG1 and beginning of TOG2 as K243 and L319, respectively. A linker for a given protein was measured counting these boundary residues and all residues in between. To reduce 'overcounting' artifacts, before making the histogram we thinned the set of sequences using the 'remove redundancy' (threshold = 0.99) feature of Jalview.

## Kinetic model

We wrote a computer program to perform kinetic Monte Carlo simulations of microtubule elongation. We implemented an algorithm similar to one described previously (*VanBuren et al., 2002*). In the program, the microtubule lattice is represented by a two dimensional lattice with a staggered periodic boundary condition to mimic the cylindrical microtubule structure. A 5 × 13 section of the lattice was designated as a 'seed' and considered to be permanently occupied. The program simulates microtubule elongation one biochemical reaction (in this case subunit addition or dissociation, GTP hydrolysis is ignored) at a time. A lattice site is considered available for subunit addition if it is empty and a neighboring site is occupied. The rate of subunit addition into any available site is given by k$_{on}$*[αβ-tubulin]. Occupied sites (excepting the seed) are considered available for dissociation, with the rate of

dissociation given by $k_{on}*K_D$ where $K_D$ is the affinity of interaction, which is determined by the neighbor state (number and type of lattice contacts) and obtained from longitudinal and corner affinities through thermodynamic coupling (*Erickson and Pantaloni, 1981*; *VanBuren et al., 2002*). Simulations begin with only 13 possible events, each on an association onto the end of a protofilament. Possible events are stored in an indexed priority queue, sorted by their 'execution times' (*Gibson and Bruck, 2000*). Execution times are determined by first sampling a random number x between 0 and 1, and then calculating the time as $-(1/rate)*ln(x)$, where *rate* gives the appropriate first- or pseudo-first order rate constant. At each step the event with the shortest execution time is implemented, the simulation time is advanced accordingly, and the list of possible events and their associated rates is updated to account for changes in subunit neighbor state. To obtain the length (in µm) of a simulated microtubule at a given time, we divide the number of subunits by 1625, the number of αβ-tubulin subunits in 1 µm of microtubule.

To model our proposed mechanism for polymerase action, we accelerated the rate at which unpolymerized subunits add adjacent to longitudinally associated 'singletons' at the microtubule end (or next to more tightly bound 'corner' subunits) (*Figure 7*). This acceleration mimics a 'tethering' effect. Control experiments showed that accelerating the rate for adding on top of a longitudinally-associated singleton had negligible effect (not shown). To keep the model as simple as possible, we did not account for any potential stabilization of TOG-bound singletons at the microtubule end.

## Acknowledgements

We thank D Borek for assistance with diffraction data processing, D Tomchick and Z Chen in the UT Southwestern Structural Biology Core facility for help and advice. H Yu, D Borek, X Zhang, and E Goldsmith gave comments on the manuscript. LMR is the Thomas O Hicks Scholar in Medical Research. Results shown in this report are derived from work performed at Argonne National Laboratory, Structural Biology Center at the Advanced Photon Source, beamline 19-ID. Argonne is operated by UChicago Argonne, LLC, for the US Department of Energy, Office of Biological and Environmental Research under contract DE-AC02-06CH11357. F-AP and EG were supported by NIH T32 GM008297, and ESV was supported by an NSF predoctoral fellowship. This work was supported by grants NIH GM098543, and NSF MCB1054947. Coordinates have been deposited in the Protein Data Bank, accession code 4U3J.

## Additional information

### Funding

| Funder | Grant reference number | Author |
| --- | --- | --- |
| National Science Foundation | MCB1054947 | Luke M Rice |
| National Institute of General Medical Sciences | T32 GM008297 | Elisabeth A Geyer, Felipe-Andrés Piedra |
| National Institute of General Medical Sciences | GM053163 | Zbyszek Otwinowski |
| National Science Foundation | Predoctoral Fellowship | Emily S Vu |
| Howard Hughes Medical Institute | HHMI | Nick V Grishin |
| National Institute of General Medical Sciences | GM098543 | Luke M Rice |

The funders had no role in study design, data collection and interpretation, or the decision to submit the work for publication.

### Author contributions

PA, EAG, RB, Acquisition of data, Analysis and interpretation of data, Drafting or revising the article; SM, F-AP, ESV, Acquisition of data, Analysis and interpretation of data; ZO, NVG, Conception and design, Analysis and interpretation of data; CAB, Conception and design, Analysis and interpretation of data, Drafting or revising the article; LMR, Conception and design, Acquisition of data, Analysis

and interpretation of data, Drafting or revising the article, Contributed unpublished essential data or reagents

# Additional files

## Major datasets

The following dataset was generated:

| Author(s) | Year | Dataset title | Dataset ID and/or URL | Database, license, and accessibility information |
|---|---|---|---|---|
| Ayaz P, Rice LM | 2014 | TOG2:alpha/beta-tubulin complex | http://www.pdb.org/pdb/explore/explore.do?structureId=4U3J | Publicly available at RCSB Protein Data Bank. |

The following previously published datasets were used:

| Author(s) | Year | Dataset title | Dataset ID and/or URL | Database, license, and accessibility information |
|---|---|---|---|---|
| Ayaz P, Ye X, Huddleston P, Brautigam CA, Rice LM | 2012 | A TOG:αβ-tubulin complex structure reveals conformation-based mechanisms for a microtubule polymerase | http://www.pdb.org/pdb/explore/explore.do?structureId=4FFB | Publicly available at RCSB Protein Data Bank. |
| Ravelli RB, Gigant B, Curmi PA, Jourdain I, Lachkar S, Sobel A, Knossow M | 2004 | Tubulin-colchicine: stathmin-like domain complextubulin-colchicine: stathmin-like domain complex | http://www.pdb.org/pdb/explore/explore.do?structureId=1SA0 | Publicly available at RCSB Protein Data Bank. |
| Slep KC, Vale RD | 2007 | Structural basis of microtubule plus end tracking by XMAP215, CLIP-170, and EB1 | http://www.pdb.org/pdb/explore.do?structureId=2qk1 | Publicly available at RCSB Protein Data Bank. |
| Lowe J, Li H, Downing KH, Nogales E | 2001 | Refined structure of alpha-beta tubulin from zinc-induced sheets stabilized with taxol | http://www.pdb.org/pdb/explore/explore.do?structureId=1JFF | Publicly available at RCSB Protein Data Bank. |

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
