## [Decision Letter]

Thank you for sending your work entitled “A tethered delivery mechanism explains the catalytic action of a microtubule polymerase” for consideration at *eLife.* Your article has been favorably evaluated by Randy Schekman (Senior editor) and 4 reviewers, one of whom, Steve Harrison, is a member of our Board of Reviewing Editors, and one of whom, William Hancock, has agreed to reveal his identity.

The Reviewing editor and the other reviewers discussed their comments before we reached this decision, and the Reviewing editor has assembled the following comments to help you prepare a revised submission.

This manuscript reports the structure of αβ-tubulin-bound Stu2 TOG2 and shows that the tubulin heterodimer is in the “curved” configuration, like the tubulin bound to TOG1. Further experiments show that tubulin binding to two linked domains is not cooperative. A series of experiments with “designed” Stu2 fragments then shows that any two TOGs on one molecule (either monomer or dimer) will give nearly normal catalytic activity. The authors find a minimum linker length for full activity of about 40 residues (consistent with the ca.300 TOG-TOG sequences in the database); the linker sequence is irrelevant. The manuscript then goes on to suggest a specific model for polymerization and to report results of simulations that embody the model. The notion is that one of the two required TOGs binds an αβ-tubulin that has associated longitudinally with the end of a MT, rather than at a “corner”, where the helical array creates lateral as well as longitudinal contacts. The former contact is, of course, weaker than the latter, but the model postulates that the high local concentration of αβ-tubulin created by the relatively weakly bound heterodimer on the second TOG domain (or presumably on any of the additional ones in a dimer or in a many-TOG monomer) promotes addition adjacent to the first heterodimer. Lateral contacts then enforce a straight conformation for both, releasing the catalyst and stabilizing the added pair as part of the growing end.

The reviewers agreed that the experimental data are sound and that they impose interesting constraints on the mechanism of polymerization. We invite a revision that attends to the following questions and reservations.

1) The authors used HYDROPRO for the prediction of the sedimentation coefficients of the extended and laterally and longitudinally associated tubulin subunits with Stu2. As this information was used to interpret the AUC binding experiments of TOG1-TOG2 with tubulin using different assumption about cooperativity, the authors should provide more details about the exact way these calculations were performed. In this context, it is a bit surprising that the sedimentation coefficients for the two putative laterally and longitudinally arranged tubulin subunits in complexes with Stu2 are similar. Do the authors have an explanation for this similarity?

2) Modeling. The reviewers found the explanation of the rationale for the model – and even its specific details – hard to follow, although there was agreement that the model is interesting and plausible enough to be included. Because the data will stand, even if future work alters the model, the authors might want consider moving the entire model description and outcome into the Discussion, or in some other way underscore the distinction. A revision of the manuscript should in any case respond to the following points:

a) The authors considered only one possible mode of action, in which one TOG domain of Stu2 binds to a tubulin subunit at the distal end of a microtubule while the other TOG domain binds to soluble tubulin. Moreover, they did not consider the role of the basic microtubule binding domain, which is well known to be crucial for the XMAP215 polymerase activity. They should explain why they simplified their system to such a degree and why they did not consider additional possibilities.

b) The model assumes that when a singleton tubulin adds, Stu2p instantaneously binds to it. Brouhard et al. measured D=0.3 µm^2^/s for XMAP215, which means it would take 13 µsec to diffuse 8 nm. That seems pretty fast. But singletons also dissociate very rapidly. For k_on_=4 x 10^6^ M^-1^s^-1^ and 8 mM K_D_ of longitudinal affinity used by Ayez, k_off_ = 3.2 x 10^4^ s^-1^, meaning that singletons have a 30 µsec residence time if not stabilized. So it is likely that some singletons will not have a bound Stu2p, and association is not actually instantaneous. This effect will not fundamentally change the model performance – it is one of degree, and it will be diminished with stronger longitudinal associations simulated in Figure 7 – but the authors should comment.

c) The model assumes that the second TOG domain will come with an attached tubulin. If the model were that one TOG domain binds the singleton tubulin and then the second binds a free tubulin, that would be a problem – even with a 10^7^ M^-1^s^-1^ on-rate and 1 µM tubulin concentration, the wait would be 100 µsec (longer than the association time of the singleton). Earlier in the paper, the authors find the Kd for tubulin binding to the TOG domains to be 100nM. That means that at 1µM free tubulin, 90% of TOG domains will have tubulin bound. So the assumption that the second TOG comes with a bound tubulin seems valid. One corollary, however, is that with µM free tubulin concentrations in solution, the TOG domains should all have a bound tubulin when the Stu2p is diffusing on the microtubule, held there by its basic domain, and the singleton will also be delivered by the Stu2p in most cases. It would have to be, because less than 10% of the Stu2p diffusing on the microtubule will have a free site that can bind the singleton. This point doesn't really change the model behavior, but it does give a picture of tubulin-bound TOGs diffusing on the microtubule that isn't conveyed in the paper. (See also comment a above).

d) The corner model, which is not consistent with the data, is a nice control. It was, however, difficult to clarify the differences between the two models – in both of them there is a “corner effect”, but one is with a curved singleton and the second is with a straight and stabilized subunit. The authors could do some work to clarify this difference.

e) The y-axis of the growth rate plots in Figure 7 are in units µm/µM-min. Unless the reviewers are missing something, the units should be µm/min. When measuring the slope, the concentration comes in, but that's a different calculation.

---

## [Author Response]

*1) The authors used HYDROPRO for the prediction of the sedimentation coefficients of the extended and laterally and longitudinally associated tubulin subunits with Stu2. As this information was used to interpret the AUC binding experiments of TOG1-TOG2 with tubulin using different assumption about cooperativity*, *the authors should provide more details about the exact way these calculations were performed. In this context, it is a bit surprising that the sedimentation coefficients for the two putative laterally and longitudinally arranged tubulin subunits in complexes with Stu2 are similar. Do the authors have an explanation for this similarity?*

We were not sufficiently clear about the role of the HYDROPRO calculations, which are not necessary for the conclusion that TOG1-TOG2 binds non-cooperatively to two αβ-tubulins. The argument for this is as follows: if there is (say) 10X positive cooperativity, then under the conditions of Figure 3 nearly all of the TOG1-TOG2 would be bound to two αβ-tubulins, and accordingly the species sedimenting at 9.1 S would be this ‘two tubulin’ complex. If there is less cooperativity, then there would be less two tubulin complex, and the observed species at 9.1 S would represent a ‘reaction boundary’ that contains contributions from tubulin, TOG1-TOG2, the one tubulin complex (which sediments at 7.1 S, Figure 3) and the two tubulin complex (which now has to sediment at greater than 9.1 S, otherwise the average would be lower than what we observed in Figure 3). The lower boundary for each ‘swath’ in Figure 3 corresponds to assuming the two tubulin complex sediments at its lower bound of 9.1 S. Only the non-cooperative swath has a lower boundary that is close to the measured values.

We provided the HYDROPRO values to provide some context for how 9.1 S might compare to different possible compact and extended configurations of the two tubulin complex, to obtain an upper bound for the sedimentation coefficient for compact complexes, and to get a sense of the robustness of our conclusion. In our initial submission we had used PyMOL to prepare coordinate files with different separations between TOG1:αβ-tubulin and TOG2:αβ-tubulin complexes, without explicitly including the linker (for which we did not have coordinates, obviously).

In the revised submission we re-visited these calculations after using MODELLER to generate linker coordinates (5 different sets of linker coordinates were generated for each model to provide a sense of how sensitive the predictions were to the modeling). The predicted sedimentation coefficients of these more complete models are somewhat lower than what we reported in our initial report (e.g. 10.5 S for longitudinal model with linker compared to 11.1 S for longitudinal model without linker), but the overall conclusion remains unchanged: the observed sedimentation coefficient of 9.1 S supports a more extended model and the titration data do not show evidence of significant positive or negative cooperativity for forming the TOG1-TOG2:(αβ-tubulin)_2_ complex.

We have updated the main text, Figure 3 and its legend and the Methods section to clarify how we did these calculations and how they contribute to the conclusion we drew.

The similarity of the calculated sedimentation coefficients for longitudinal vs lateral compact complexes took some of us by surprise also. This can be understood as follows. Assuming the same partial specific volume and solvent density, the sedimentation coefficient is proportional to M/f, where M is the molar mass and f is the frictional coefficient. To obtain significantly different sedimentation coefficients for the same mass requires large changes in f. For a back of the envelope calculation, the longitudinal and lateral complexes can be approximated as prolate ellipsoids with axial ratios of 2.1 and 1.43, respectively. These axial ratios lead to frictional ratios (assuming 0.3 g/g hydration) of 1.17 and 1.14. Thus, HYDROPRO calculated relatively similar values from the two configurations we considered because they have relatively similar frictional ratios and the same molar mass.

*2) Modeling. The reviewers found the explanation of the rationale for the model – and even its specific details – hard to follow, although there was agreement that the model is interesting and plausible enough to be included. Because the data will stand, even if future work alters the model, the authors might want consider moving the entire model description and outcome into the discussion, or in some other way underscore the distinction*. *A revision of the manuscript should in any case respond to the following points:*

We are grateful for the chance to clarify the modeling, and we have modified the text to better explain our decision-making and to also address the specific points raised below.

*a) The authors considered only one possible mode of action, in which one TOG domain of Stu2 binds to a tubulin subunit at the distal end of a microtubule while the other TOG domain binds to soluble tubulin. Moreover, they did not consider the role of the basic microtubule binding domain, which is well known to be crucial for the XMAP215 polymerase activity. They should explain why they simplified their system to such a degree and why they did not consider additional possibilities*.

We did not mean to give the impression that we were neglecting/ignoring the basic domain. In our understanding without a basic domain (or some other equivalent mechanism for microtubule association) the TOGs would not function as polymerases and would instead be αβ-tubulin sequestering factors. However, to explicitly model the diffusive process, and how being on the microtubule might modulate the affinity and/or kinetics of TOG:tubulin interactions, seemed too complicated at this time (especially given the relatively limited data available to restrain the additional parameters that would be required). We have now added a few sentences to better clarify our view of the basic domain and why it does not figure explicitly into our model.

Why did we simplify the system so much? It seemed clear to us that tethering-induced increases in the apparent concentration of unpolymerized tubulin could enhance growth rates. But it was not obvious that tethering near singletons would produce a catalytic effect. Thus, we implemented the simplest possible representation of the tethering effect to focus on this one narrow question. We cannot exclude other mechanisms (e.g. a diffusing polymerase carrying one or more tubulins might shuttle them to the end, see below for more on this), but the processive action demonstrated for XMAP215 by Brouhard et al. suggests that shuttling does not play a dominant role. In essence, we used the modeling as a device to synthesize our structural and biochemical observations into a kinetic framework capable of addressing the question of catalytic action in a way that structures and binding constants alone cannot. We have tried to make the rationale and limitations clearer through text modifications.

An obvious next step for us will be to perform in vitro reconstitutions using re-engineered variants. In this way we hope to obtain data that will support more parameter-intensive modeling of more aspects of the polymerase function.

*b) The model assumes that when a singleton tubulin adds, Stu2p instantaneously binds to it. Brouhard et al. measured D=0.3 µm*^*2*^*/s for XMAP215, which means it would take 13 µsec to diffuse 8 nm. That seems pretty fast. But singletons also dissociate very rapidly. For k*_*on*_*=4 x 10*^*6*^
*M*^*-1*^*s*^*-1*^
*and 8 mM K*_*D*_
*of longitudinal affinity used by Ayez, k*_*off*_
*= 3.2 x 10*^*4*^
*s*^*-1*^*, meaning that singletons have a 30 µsec residence time if not stabilized. So it is likely that some singletons will not have a bound Stu2p, and association is not actually instantaneous. This effect will not fundamentally change the model performance – it is one of degree, and it will be diminished with stronger longitudinal associations simulated in*
Figure 7*, but the authors should comment*.

This is a good comment that identifies a compromise inherent to our minimalist approach. We had included language attempting to make this compromise clear, and we have now added additional emphasis to the fact that in reality the maximal polymerase activity will likely be determined by its degree of saturation at the microtubule end. Varying the ‘tethered association rate’ is an artificial device that we use to avoid having to explicitly model the question of polymerase saturation at the microtubule end.

*c) The model assumes that the second TOG domain will come with an attached tubulin. If the model were that one TOG domain binds the singleton tubulin and then the second binds a free tubulin, that would be a problem – even with a 10*^*7*^
*M*^*-1*^*s*^*-1*^
*on-rate and 1 µM tubulin concentration, the wait would be 100 µsec (longer than the association time of the singleton). Earlier in the paper, the authors find the Kd for tubulin binding to the TOG domains to be 100nM. That means that at 1µM free tubulin, 90% of TOG domains will have tubulin bound. So the assumption that the second TOG comes with a bound tubulin seems valid. One corollary, however, is that with µM free tubulin concentrations in solution, the TOG domains should all have a bound tubulin when the Stu2p is diffusing on the microtubule, held there by its basic domain, and the singleton will also be delivered by the Stu2p in most cases. It would have to be, because less than 10% of the Stu2p diffusing on the microtubule will have a free site that can bind the singleton. This point doesn't really change the model behavior, but it does give a picture of tubulin-bound TOGs diffusing on the microtubule that isn't conveyed in the paper. (See also comment a above)*.

We hope that our responses to the previous two comments also substantially address this one by better conveying the picture of tubulin-bound TOGs diffusing on the microtubule. It is certainly possible that a polymerase diffuses to the microtubule end carrying one or more tubulins. The Brouhard et al. paper indicates that for XMAP215, at least, this shuttling function cannot explain the observed increase in growth rate. Thus, some form of ‘end-resident’ polymerase must make substantial contributions to activity, and this is what our model is attempting to capture.

*d) The corner model, which is not consistent with the data, is a nice control. It was, however, difficult to clarify the differences between the two models – in both of them there is a “corner effect”, but one is with a curved singleton and the second is with a straight and stabilized subunit. The authors could do some work to clarify this difference*.

This was a good way of explaining a potential source of confusion. It is correct that in each case the incoming tubulin adds into ‘corner’ sites that differ because curved singletons are much less tightly bound to the microtubule end than straight subunits that already participate in their own corner interactions. The lifetime of the stably bound tubulins (‘straight corner’) is long compared to the tethered delivery time, so every delivery to these sites will be ‘productive’ and accordingly a substantial enhancement in growth rate is predicted. In contrast, the relatively short lifetime of singletons means that they tend to dissociate faster than the tethered delivery time. We have altered the text in a few places to make this distinction clearer.

*e) The y-axis of the growth rate plots in*
Figure 7
*are in units µm/µM-min. Unless the reviewers are missing something, the units should be µm/min. When measuring the slope, the concentration comes in, but that's a different calculation*.

The reviewers were not missing anything, this was a copy/paste error that slipped through the cracks. The units are now correct in Figure 7 and its supplement.